# A Chymotrypsin-Dependent Live-Attenuated Influenza Vaccine Provides Protective Immunity against Homologous and Heterologous Viruses

**DOI:** 10.3390/vaccines12050512

**Published:** 2024-05-08

**Authors:** Peiqing He, Mengxuan Gui, Tian Chen, Yue Zeng, Congjie Chen, Zhen Lu, Ningshao Xia, Guosong Wang, Yixin Chen

**Affiliations:** State Key Laboratory of Vaccines for Infectious Diseases, National Institute of Diagnostics and Vaccine Development in Infectious Diseases, State Key Laboratory of Molecular Vaccinology and Molecular Diagnostics, School of Life Sciences, School of Public Health, Xiamen University, Xiamen 361102, China; hepeiqing2021@stu.xmu.edu.cn (P.H.); guimengxuan@stu.xmu.edu.cn (M.G.); chentian28993@163.com (T.C.); zengyue9914@163.com (Y.Z.); chencongjie@stu.xmu.edu.cn (C.C.); luj04406@xmu.edu.cn (Z.L.);

**Keywords:** influenza, live-attenuated vaccine, mucosal immunology, broad-spectrum protection, cleavage site

## Abstract

Influenza virus is one of the main pathogens causing respiratory diseases in humans. Vaccines are the most effective ways to prevent viral diseases. However, the limited protective efficacy of current influenza vaccines highlights the importance of novel, safe, and effective universal influenza vaccines. With the progress of the COVID-19 pandemic, live-attenuated vaccines delivered through respiratory mucosa have shown robustly protective efficacy. How to obtain a safe and effective live-attenuated vaccine has become a major challenge. Herein, using the influenza virus as a model, we have established a strategy to quickly obtain a live-attenuated vaccine by mutating the cleavage site of the influenza virus. This mutated influenza virus can be specifically cleaved by chymotrypsin. It has similar biological characteristics to the original strain in vitro, but the safety is improved by at least 100 times in mice. It can effectively protect against lethal doses of both homologous H1N1 and heterologous H5N1 viruses post mucosal administration, confirming that the vaccine generated by this strategy has good safety and broad-spectrum protective activities. Therefore, this study can provide valuable insights for the development of attenuated vaccines for respiratory viruses or other viruses with cleavage sites.

## 1. Introduction

According to WHO statistics, about 1 billion cases of influenza occur each year, including 3–5 million severe cases, resulting in 290,000–650,000 respiratory deaths each year [1]. Influenza A is one of the main factors contributing to the burden of these diseases [2]. It belongs to the *Orthomyxoviridae* family, and its genome is composed of 8 segments of single-stranded RNA, which can encode at least 11 proteins including the major surface proteins hemagglutinin (HA) and neuraminidase (NA) [3,4]. Based on their antigenic differences, there are 19 types of HA and 11 types of NA [5,6]. H1N1, H5N1, and H7N9 have high pathogenicity in humans and have led to global pandemics [7]. The H1N1 virus, in 1918, also known as the Spanish Flu, caused 50 million deaths worldwide [8], while the swine influenza H1N1 in 2009 resulted in 200,000 deaths globally in the first 12 months and continues to circulate as a seasonal flu [9]. As of 21 April 2023, 874 cases of human infection with H5N1 have been reported in 23 countries, with 458 (52.4%) deaths [10]. Additionally, more than 75 cases of cross-species H5N6 infections in humans have been reported in recent years, seemingly exhibiting greater virulence than H5N1 [11]. Facing these challenges, the development of highly efficient vaccines and antiviral drugs against influenza A virus holds significant importance.

Vaccine is one of the most effective means of preventing viral infections. Influenza vaccines on the market are mainly inactivated vaccines, administered via intramuscular injection to induce a systemic immune response. However, due to the antigenic drift of the influenza virus, despite annual updates to the vaccine strains, the protective effectiveness of the influenza vaccine remains relatively low [12]. Hence, the development of broad-spectrum and highly efficient influenza vaccines to effectively combat virus mutations is crucial. Although scientists have designed and attempted various broad-spectrum vaccines, such as HA stem vaccines and nanoparticle vaccines, their safety and efficacy in populations still require further validation [13,14,15]. Therefore, it is necessary to explore more strategies for the development of broad-spectrum vaccines. With the outbreak of the SARS-CoV-2 pandemic [16], a considerable amount of scientific research has indicated that mucosal vaccines for the respiratory tract are among the most effective means of defending against respiratory virus infections [17,18,19,20,21]. Particularly, live-attenuated vaccines have been shown to induce a variety of protective immune responses, such as humoral immunity, cellular immunity, innate immunity, and trained immunity [20], thus effectively preventing the infection and the occurrence of related diseases.

Reducing the virulence of viruses is a key factor in the development of live-attenuated vaccines. In previous studies, the continuous viral adaptive passage [22], truncation and deletion of the NS1 gene [23], and cold adaptation modifications have all been effective in reducing the virulence of viruses [24]. It is noteworthy that these modified attenuated vaccines still replicate and proliferate in vivo. Due to the high variability of the influenza virus, the virus can acquire certain virulence-restoring mutations during multiple rounds of replication and propagation, leading to the restoration of viral virulence. Therefore, the safety of attenuated vaccine strains is an issue that cannot be ignored. The HA protein of the influenza virus plays a very important role in the infection of virus. It binds to the sialic acid receptor on cells through the receptor-binding domain. The Arg site on the cleavage site of HA can be cleaved by respiratory tract-expressed HAT (human airway trypsin-like protease) and TMPRSS2 (transmembrane protease, serine S1 member 2), and HA-mediated membrane fusion releases the genome into the cell, completing the virus infection [25,26]. It is clear that the cleavage of the HA cleavage site is a necessary condition for the influenza virus infection.

In this study, by mutating the Arg on the cleavage site, we constructed a chymotrypsin-dependent influenza virus strain that exhibited similar biological activity to its parent strain in vitro, but due to its inability to be activated by HAT and TMPRSS2, it showed good attenuating activity, and its safety was at least 100 times higher than that of the parent strain in the mouse model. When compared with inactivated virus immunization, the live-attenuated vaccine induced a stronger humoral immune response through mucosal immunity and provided 100% protection against lethal doses of both the homologous H1N1 and the heterologous H5N1, demonstrating safe and highly effective broad-spectrum protective activity. Therefore, this influenza virus attenuation strategy provides good guidance for the development of attenuated broad-spectrum vaccines for respiratory mucosal immunity. 

## 2. Materials and Methods

### 2.1. Cell Lines

The Madin–Darby canine kidney (MDCK) cell line was generously gifted by Professor Honglin Chen from the University of Hong Kong. The human embryonic kidney 293FT cell line was purchased from Invitrogen (R700-07). Cells were cultured in Dulbecco’s Modified Eagle Medium (DMEM) (D5796, Sigma-Aldrich, St. Louis, MO, USA), supplemented with 10% low-endotoxin fetal bovine serum (FBS) (A0500-3011, Cegrogen, Ebsdorfergrund, Germany) and a penicillin–streptomycin mixture. 

### 2.2. Plasmids and Proteins

The HA gene of A/California/04/2009 (H1N1) was cloned into the phw2000 vector. Primers to mutate the arginine (Arg) at position 344 to phenylalanine (Phe), tyrosine (Tyr), or tryptophan (Trp) were designed. The primer sequence was as follows:
CA4-R344F-FCCGTCTATTCAATCTTTCGGCCTATTTGGGGCCATTG37 bpCA4-R344F-RCCCCAAATAGGCCGAAAGATTGAATAGACGGGATATTCCTCAATC45 bpCA4-R344W-FCCGTCTATTCAATCTTGGGGCCTATTTGGGGCCATTG37 bpCA4-R344W-RCCCCAAATAGGCCCCAAGATTGAATAGACGGGATATTCCTCAATC45 bpCA4-R344Y-FCCGTCTATTCAATCTTACGGCCTATTTGGGGCCATTG37 bpCA4-R344Y-RCCCCAAATAGGCCGTAAGATTGAATAGACGGGATATTCCTCAATC45 bp

After obtaining positive clones, sequence the DNA in the plasmid to ensure that the point mutation has been successfully introduced. The codon-optimized gene sequences for the A/California/04/2009 (H1N1) and A/Vietnam/1194/2004 (H5N1) HA proteins that were capable of expressing in the extracellular region, along with an 8× His-tag, were cloned into the pTT5 vector (Shanghai Sangon Biotechnology Co., Ltd., Shanghai, China). Proteins were expressed using the 293 Expression System, and the supernatant was purified using nickel column affinity chromatography according to the manufacturer’s instructions (5893801001, Roche, Basel, Switzerland).

### 2.3. Generation and Passage of Influenza Viruses

The eight genes of A/California/04/2009 (H1N1) (GenBank No. MN371610.1-371617.1) were cloned into the phw2000 vector, as described previously [21], and the plasmids were transfected into 293FT cells. Eight hours later, the medium was changed to remove the DNA mixture. After 24 h, a final concentration of 5 μg/mL α-chymotrypsin (C3142, Sigma-Aldrich), 2 mmol/L CaCl_2_, 1 mmol/L HCl, or 2.5 μg/mL TPCK-treated trypsin (Sigma, St. Louis, MO, USA) was added to the cells in Opti-MEM medium (31985-070, Gibco, Billings, MT, USA). Forty-eight hours later, the supernatant was collected and used to infect MDCK cells to observe cytopathic effects. The viral titer was determined using the plaque assay with MDCK cells.

### 2.4. Immunofluorescence

MDCK cells were infected with different dilutions of the influenza virus. After 24 h, cells were washed once with a phosphate buffer saline (PBS) solution, then fixed with paraformaldehyde for 15 min. Following another wash with the PBS solution, cells were permeabilized with a PBST (0.3% Trition X100 in PBS) solution for 15 min and then washed again with PBS. Cells were blocked with goat serum at 37 °C for 2 h. The goat serum was discarded, and cells were incubated with appropriately diluted fluorescently labeled antibodies at 37 °C for 30 min. After discarding the supernatant, cells were washed 5 times with the PBS solution, stained with a DAPI solution (D1306, Invitrogen, Waltham, MA, USA) for 5 min to label nuclei, and then washed 5 times with PBS. Images were captured using a high-content imaging system (Operetta CLS).

### 2.5. ELISA

ELISA were carried out as described previously [27]. Microtiter plates (Wantai BioPharm, Beijing, China) were coated overnight at 4 °C with 100 ng of recombinant A/California/04/2009 (H1N1) and A/Vietnam/1194/2004 (H5N1) HA protein. The plates were washed three times with PBS containing 0.1% *v*/*v* Tween-20 and blocked with a blocking solution (containing 2% sucrose, 0.2% sodium caseinate, and 2% gelatin in phosphate-buffered saline) at 37 °C for 2 h. The plates were then washed with the wash buffer. A series of tenfold diluted immune sera were added to the wells and incubated at 37 °C for 30 min. After washing five times, 100 μL of horseradish peroxidase (HRP)-conjugated goat anti-mouse antibody solution was added to each well and incubated at 37 °C for 30 min. Following five washes, a substrate solution (Wantai BioPharm, Beijing, China) was added. The reaction was stopped after 15 min by 2 M H_2_SO_4_. Absorbance was measured at 450 nm.

### 2.6. Plaque Assay

MDCK cells were seeded in a 6-well plate and allowed to reach confluence. The supernatant was discarded, and the cells were washed once with the PBS solution. A serum-free DMEM medium was added before 10-fold serially diluted influenza virus samples were introduced into the wells. After thorough mixing, the plate was incubated in a 37 °C incubator. One hour later, the supernatant was discarded and replaced with 0.75% agarose in the respective culture medium to restrict the spread of free virus. The plate was then incubated in a 37 °C CO_2_ incubator for 48 h. Cells were fixed with a formaldehyde solution and stained with a crystal violet solution. Plaques were counted, averaged and multiplied by the dilution factor to determine viral titer as PFU/mL.

### 2.7. Ethics Statements

All animals involved in this study were housed and cared for in an Association for the Assessment and Accreditation of Laboratory Animal Care (AAALAC)-accredited facility. All experimental procedures with mice were conducted according to Chinese animal use guidelines and were approved by the Institutional Animal Care and Use Committee (IACUC) of Xiamen University (XMULAC20200232).

### 2.8. Animal Experiments

To assess the safety of CA04 and CA04-F, seventy 6-week-old female BALB/C mice were purchased from the Shanghai Slake Laboratory Animal Co., Ltd. (Shanghai, China). The mice were randomly divided into 14 groups, with 5 mice per group to assess the safety of the influenza viruses. A/California/04/2009-F mutation (CA04-F) and A/California/04/2009 (CA04), each with a titer of 10^6^ pfu/mL, were diluted to 10^5^, 10^4^, 10^3^, 10^2^, and 10^1^ pfu/mL. The mice were anesthetized using isoflurane in an anesthesia machine. Subsequently, 6 different titers of CA04 and CA04-F viruses (50 μL) were administered into the nasal cavity of each mouse to form 12 groups. The mice of the other 2 groups administered PBS (50 μL) into the nasal cavity as a control. Adequate food, water, and appropriate breeding conditions were provided. The weight and survival rate of the mice were monitored daily. For the purpose of calculating survival rates, a mouse was considered dead when its weight decreased by 25% from its initial weight.

To assess the protective activity of homologous and heterologous viruses after CA04-F mucosal immunization and inactivated CA04 administered via intramuscular injection, seventy-eight 6-week-old female BALB/C mice were purchased from the Shanghai Slake Laboratory Animal Co., Ltd. and were randomly divided into three groups, with 26 mice per group. For the first group, the mice were anesthetized using isoflurane in an anesthesia machine, and then 50 µL of diluted CA04-F virus at a concentration of 10^6^ pfu/mL were instilled into the nasal cavity of each mouse. A second immunization was carried out 14 days later. The second group was immunized via intramuscular injection with the same titer of inactivated CA04 virus mixed with aluminum adjuvant. A booster immunization was administered 14 days later. The third group received nasal immunization with PBS solution and muscular immunization with PBS mixed with aluminum adjuvant. Thirteen mice from each group were challenged with 5000 pfu A/California/04/2009 (H1N1) and A/Vietnam/1194/2004 (H5N1) viruses to assess the efficacy of the immunization protocols. Five mice from each group were monitored for weight and survival rate, while the other mice underwent lung harvest on day 3 and 5 post-challenge. The collected lung tissues were analyzed for viral titers.

### 2.9. Quantitative Reverse Transcriptase PCR (qRT-PCR) 

Total RNA from lung tissues was extracted using the Viral RNA/DNA Extraction kit (NA007, Genmagbio, Changzhou, China) according to the manufacturer’s protocol and then converted to cDNA by QIAGEN OneStep RT-PCR Kit (210212, Qiagen, Hilden, Germany). The cDNA was then subjected to qRT-PCR using primers and probes that are specific to the HA gene. The sequences of the primers and probes were as follows:
H1HA-FGCATAACGGGAAACTATGCAA21 bpH1HA-RGCTTGCTGTGGAGAGTGATTC21 bpH1HA-probeFAM-TTACCCAAATGCAATGGGGCTACCCC-BBQ26 bp

qRT-PCR was performed under the following reaction conditions: 50 °C for 15 min, 95 °C for 15 min, followed by 45 cycles of 94 °C for 15 s, 55 °C for 60 s, and lastly 35 °C for 30 s.

### 2.10. HA Assay

An HA assay was carried out as described previously [28]. Briefly, influenza samples were serially twofold diluted and mixed with an equal volume of 0.75% guinea pig red blood cell suspension that was added to the wells, then incubated for 1 h at room temperature. The lowest concentration of the sample that completely formed hemagglutination was designated the hemagglutination unit (HAU).

### 2.11. HAI Assay

Using a receptor-destroying enzyme (Denka-Seiken, Tokyo, Japan), the treated immune serum was diluted in a twofold gradient and mixed with influenza A/California/04/2009 (H1N1) or A/Vietnam/1194/2004 (H5N1) viruses at 8 hemagglutination units in equal volumes. After incubating at room temperature for 1 h, an equal volume of 0.75% guinea pig red blood cell suspension was added. Following another hour of incubation at room temperature, the lowest concentration of the serum that could inhibit hemagglutination was considered as the serum’s HAI titer.

### 2.12. Neutralization Assays

Neutralization assays were carried out as described previously [29]. MDCK cells were seeded into a 96-well plate. Heat-inactivated immunoserum was serially diluted twofold and mixed with an equal volume of virus, then incubated at 37 °C for 2 h. After incubation, 35 µL of the mixture containing 100 TCID50 of the virus were added to the MDCK cells for adsorption for 1 h. The virus supernatant was removed and replaced with DMEM-containing antibiotics and TPCK trypsin. The cells were incubated at 37 °C in a CO_2_ incubator for 72 h, and neutralization titers were determined using the HA assay. For the HA assay, 50 µL of 0.75% guinea pig red blood cell suspension were added to 50 µL of the cell culture supernatant, and incubated at room temperature for 1 h. The neutralization titer was the lowest serum concentration that was negative for hemagglutination.

### 2.13. Statistics 

Statistical significance was calculated using one-way or two-way ANOVA, as indicated in the figure legends. For all statistical analyses, differences were considered significant when the *p* value was less than or equal to 0.05 (* *p* < 0.05; ** *p* < 0.01; *** *p* < 0.001; **** *p* < 0.0001). Statistical analyses were performed using GraphPad Prism 8.

## 3. Results

### 3.1. The Rescue of a Chymotrypsin-Dependent Influenza Virus by Reverse Genetics

Chymotrypsin is a type of serine protease that is typically highly expressed in the pancreas. It can specifically cleave peptide bonds formed by the C-terminus of F, Y, W amino acids, exhibiting good specificity. To construct a chymotrypsin-dependent influenza virus, the R at the cleavage site of A/California/04/2009 (CA04) HA was mutated to an amino acid site (F, Y, W) recognizable by chymotrypsin (Figure 1A), so that the cleavage site of the mutated HA could be recognized by chymotrypsin. Utilizing the reverse genetics system of the influenza virus, the mutated HA plasmid and the other 7 plasmids of CA04 were used for virus generation; in the culture and passage of the virus, chymotrypsin replaced the original TPCK trypsin (Figure 1B). After the generated virus infected MDCK cells for 48 h, a hemagglutination assay was used to detect the titer of the influenza virus in the supernatant. The results show that hemagglutination could be detected under the cultivation conditions with chymotrypsin after mutating 344R to F and Y, with mutations to F yielding a higher titer (Figure 1C,D). In contrast, mutations to W and the negative control did not display this phenomenon (Figure 1C,D), suggesting successful influenza virus generation following mutations to F and Y.

### 3.2. The Similar Biological Characteristics between the Rescued Virus with Its Parent In Vitro

To ensure that the generated influenza virus with the mutated cleavage site still possesses efficiently infective activity, A/California/04/2009-R344F (CA04-F) and CA04 were used to infect MDCK cells. After 24 h, immunofluorescence assays were conducted to detect the expression of NP and HA in the cells. It was found that both viruses could express NP and HA in the infected cells, and both had good colocalization, confirming that CA04-F still had good infective activity (Figure 2A). To further verify the specificity of CA04-F to chymotrypsin, plaque assays were performed with both viruses under conditions with chymotrypsin or TPCK trypsin. It was discovered that CA04-F could only produce clear viral plaques under chymotrypsin conditions, while CA04 could only form plaques under TPCK trypsin conditions (Figure 2B). To explore more suitable virus culture conditions to obtain high viral titers for subsequent animal experiments, an investigation was conducted on the concentration of chymotrypsin. It was observed that the viral titers showed a normal distribution with increasing chymotrypsin concentration, reaching the highest titer at a concentration of 2.5 μg/mL (Figure 2C). This suggests that this concentration is the most suitable for the cultivation of CA04-F. 

### 3.3. The Safety of CA04-F in Mice

To evaluate the safety of CA04-F, CA04-F and CA04 were diluted at a tenfold gradient and doses were administered to mice in 50 μL volumes via nasal inhalation, monitoring changes in body weight and survival rates. The bodyweight and survival curves indicated that the mice in groups 10^6^ and 10^5^ pfu/mL of CA04 showed significant weight loss and 100% mortality (Figure 3A,B). Group 10^4^ pfu/mL experienced notable weight loss, but no deaths occurred. The remaining groups showed no significant changes. After the nasal administration of any dose of CA04-F, no significant weight changes or mortalities were observed in the mice (Figure 3C,D). This confirms the good safety profile of CA04-F, which is at least 100 times safer than CA04. Furthermore, the viral RNA levels in the lungs of the mice in groups 10^6^ pfu/mL of CA04 and CA04-F were measured using qRT-PCR on day 1, day 3, and day 5 post-infection (Figure 3E). It was found that there was no difference in viral RNA levels between the two viruses on day 1. However, on days 3 and 5, the viral RNA levels in the group infected with CA04 were significantly higher than those in the CA04-F group, indicating good attenuated virulence (Figure 3F).

### 3.4. The Humoral Immune Response Activity after Mucosal Immunization with CA04-F

To evaluate the levels of humoral immune responses after mucosal immunization with CA04-F, mice were immunized intranasally with CA04-F on day 0 and day 14, and the binding/neutralizing antibodies against the influenza virus in the mouse serum were tested on day 28. To analyze the differences in immune response between live virus mucosal immunization and traditional inactivated virus muscular immunization, a group was added that received the same dose of CA04 after heat inactivation (56 °C, 30 min), mixed with alum adjuvant, through intramuscular injection. By detecting the binding antibodies in the serum against the HA protein of CA04 and the HA protein of H5N1, it was found that the CA04-F mucosal immunization group showed stronger reactivity to both homologous and heterologous HA proteins (Figure 4A,B). Functional antibody levels in the immunized sera were tested using hemagglutination inhibition (HAI) assays and microneutralization assays, revealing that the CA04-F mucosal immunization group had higher HAI titers and neutralization titers against CA04 (Figure 4C,D). However, neither serum detected HAI activity or neutralization activity against H5N1. These data confirm that mucosal immunization with CA04-F can induce a stronger humoral immune response than the intramuscular injection of an inactivated virus, leading to the induction of higher titers of neutralizing antibodies.

### 3.5. Protective Activity of Homologous and Heterologous Viruses after CA04-F Mucosal Immunization

To evaluate the difference in protective efficacy in vivo between CA04-F mucosal immunization and inactivated CA04 administered via intramuscular injection, we challenged mice with lethal doses of A/California/04/2009 (H1N1) and A/Vietnam/1194/2004 (H5N1) via the nasal route. The body weight curves indicate that, in the homologous virus A/California/04/2009, both the CA04-F group and the CA04-Al group exhibited good protective activity against A/California/04/2009 (Figure 5A). The body weight of mice in the CA04-F group remained relatively stable without significant decline, whereas in the CA04-Al group, 1/5 of the mice experienced noticeable weight loss (Figure 5A). No deaths occurred in either group (Figure 5B). On the 3rd and 5th days post-infection, viral loads in the lung homogenates were measured. No virus was detected on either day in the CA04-F group, while the CA04-Al group displayed detectable viral loads on both days (Figure 5C). This confirmed that CA04-F mucosal immunization provided stronger protection than inactivated CA04 administered via intramuscular injection. In the heterologous H5N1 virus challenge model, the CA04-F group still exhibited robustly protective activity, with no significant weight loss or mortality among the mice, whereas in the CA04-Al group, all mice showed significant weight reduction, and 60% succumbed to the infection (Figure 5D,E). In terms of lung viral load, the CA04-F group presented only a few viruses on day 3, with no virus detected by day 5, while the CA04-Al group had lung viral loads on days 3 and 5 comparable to the control group (Figure 5F). The above data confirm that mucosal immunization with CA04-F can effectively protect against heterologous virus infections, significantly outperforming thermally inactivated CA04 transferred via intramuscular injection.

## 4. Discussion

There are various methods for attenuating influenza viruses, including passage adaptation, cold adaptation, gene deletion, or truncation. These modifications can reduce the virulence of the virus to some extent, yet it retains the ability to replicate and proliferate at low levels in vivo. Although it may not cause disease in healthy individuals, it could potentially pose a risk to immuno-compromised populations such as infants, pregnant women, the elderly, or others with immune deficiencies [30]. Given that the influenza virus is highly prone to mutations, there is a possibility that virulence-enhancing mutations could arise during replication, creating hazards. Therefore, when developing attenuated vaccines for as highly mutable viruses as influenza, it is crucial to pay particular attention to the safety and the impact of potential virulence-restoring genetic mutations. In the context of developing attenuated influenza vaccines, maintaining the virus’s ability to single-cycle infect without producing new progeny viruses capable of infecting other cells could be an important way to minimize the risk of virulence-restoring mutations. This process might be similar to self-replicating mRNA vaccines but is potentially easier to achieve while retaining the influenza virus’s affinity for certain cells or tissues [31]. Based on this rationale, researchers have made numerous attempts, such as deletions of NA [32], PB2 [33], M2 [34], etc. These modifications often require complementation in the virus-culturing cell lines, where the deleted protein is expressed, allowing the virus to replicate normally within those cells. Obtaining such gene-deleted viruses, which lack the capacity for subsequent transmission in the host due to the absence of the deleted protein, promises enhanced safety and can also serve as effective vaccine vectors for the expression of exogenous genes. However, this method requires specific cell lines engineered to express exogenous genes for culturing, which poses higher demands on vaccine production and industrialization.

The enzymatic cleavage at the influenza virus cleavage site is crucial for the virus’s ability to infect. Previous research has shown that deleting the basic amino acids at the cleavage site of the H5N1 virus, leaving only one arginine, can significantly reduce the virulence of the H5N1 virus [35], but it still may maintain a relatively high level of pathogenicity. Researchers have mutated the R in the cleavage site of A/WSN/1933 (H1N1) to Val, which becomes specifically activated by elastase, showing good protective efficacy against homologous viruses upon immunization. Elastase can cleave nonpolar amino acids, mutating this site to other amino acids, such as Alanine, also successfully generating the virus. This virus has been extensively studied and has shown good safety, with immunization inducing strong IgG, IgA, and specific IFN-γ responses, providing protection against both homologous and heterologous viruses [36]. However, elastase is highly expressed in neutrophils [37], and vaccination might lead to the activation of neutrophils expressing elastase, potentially enabling the modified virus to be activated again, gaining the ability to re-infect and replicate, which could lead to disease. Therefore, this study explored another cleavage site modification strategy, mutating the cleavage site Arg to Phe, which can specifically replicate under the conditions of chymotrypsin. This has shown good safety, and after immunization, it provides protection against both homologous and heterologous viruses, effectively suppressing viral proliferation in the lungs. This confirms that such an attenuation modification strategy is an effective way to obtain attenuated vaccines.

Cleavage sites are also present in a variety of other viruses, including coronaviruses and herpesviruses. These are mostly found on viral fusion proteins, which primarily mediate the fusion process between the virus and the cell membrane, playing a crucial role in the virus’s ability to infect. In coronaviruses, such as SARS-CoV-1 and MERS, there is only one Arg in the cleavage site, whereas SARS-CoV-2 has multiple [38]. Previous research has identified that the stronger pathogenicity of SARS-CoV-2 might be related to its cleavage site containing multiple basic amino acids. Deleting the excess basic amino acids can significantly reduce the virulence of SARS-CoV-2 [39], but it still efficiently replicates in the respiratory tract. Therefore, the modification strategy identified in this study could potentially lead to the development of a novel, attenuated pan-coronavirus vaccine that, after mucosal immunization through the respiratory tract, could provide broad-spectrum protection against various new mutant strains. Herpesviruses are a group of enveloped DNA viruses with a nucleocapsid that pose significant risks to humans and mammals [40,41]. They also offer economic benefits when used as vaccine vectors or oncolytic viruses [42,43,44]. However, effective vaccines to block viral infection have yet to be commercialized in humans. The development of herpesvirus vaccines is fraught with challenges. Herpesviruses such as PRV, BHV-1, VZV, and EBV have multiple basic amino acids at the cleavage site of their gB proteins. Research has shown that deleting basic amino acids from the cleavage site of the PRV gB protein can significantly reduce the virulence of the PRV virus, but it still retains some pathogenicity [45]. Therefore, we believe the attenuation modification strategy discovered in this article provides a new direction for the development of attenuated vaccines against a variety of viruses, including coronaviruses and herpesviruses.

Mucosal immunization might be one of the most effective methods to prevent respiratory virus infections. Traditional intramuscular injection vaccines can induce strong systemic humoral and cellular immune responses after immunization. When viruses invade, the delayed activation of the immune system provides an opportunity for the virus to infect and cause disease. However, respiratory mucosal immunity can generate robust humoral and cellular immunity at the respiratory mucosa. When viruses invade, they can be directly blocked or eliminated at the respiratory mucosa, offering stronger protective efficacy. After mucosal immunization, the CA04-F virus can produce broad-spectrum protective activity against both homologous and heterologous viruses. The efficacy of the protective mechanisms is not singular. Previous research has discovered that an attenuated influenza vaccine, lacking NS1 and carrying the gene of SARS-CoV-2 RBD (receptor binding domain), can effectively protect against both homologous and heterologous influenza viruses and SARS-CoV-2 [21]. An analysis of its protective mechanisms revealed that the vaccine induces a variety of protective immune responses at the pulmonary mucosa, including specific humoral immune responses, cellular immune responses, innate immunity, and trained immunity [20]. These play a crucial role in the vaccine’s broad-spectrum protection. Therefore, we also speculate that the CA04-F vaccine may likewise induce such multifaceted protective immune responses, effectively blocking influenza virus infection, and exerting strong, broad-spectrum protective activity.

## Figures and Tables

**Figure 1 vaccines-12-00512-f001:**
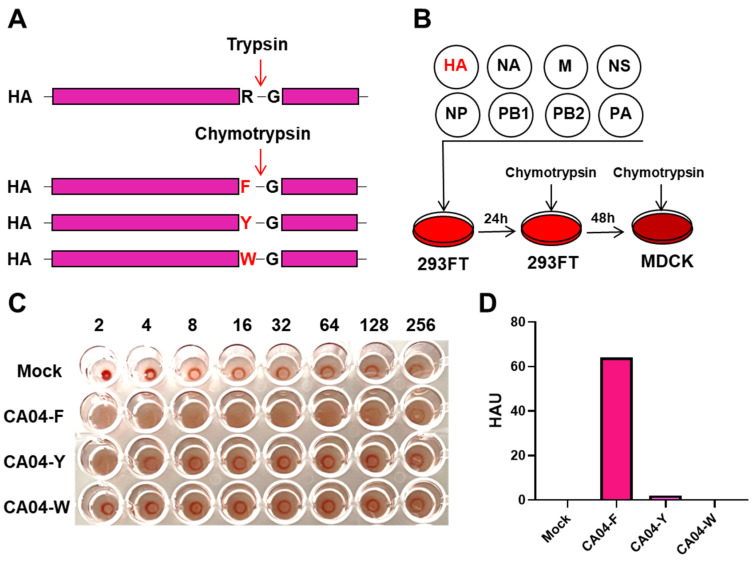
The generation of the chymotrypsin-dependent influenza virus. (**A**) Mutations in the cleavage site of the influenza virus ensure that it can potentially be cleaved by chymotrypsin. The mutated amino acids are marked in red. Hemagglutinin, HA; Arginine, R; Glycine, G; Phenylalanine, F; Tyrosine, Y; Tryptophan, W. (**B**) Flowchart of influenza virus generation based on reverse genetics. (**C**) Detection of hemagglutination activity of the generated influenza virus by HA assay. A/California/04/2009 (H1N1), CA04. (**D**) The hemagglutination unit (HAU) of the influenza virus in (**C**).

**Figure 2 vaccines-12-00512-f002:**
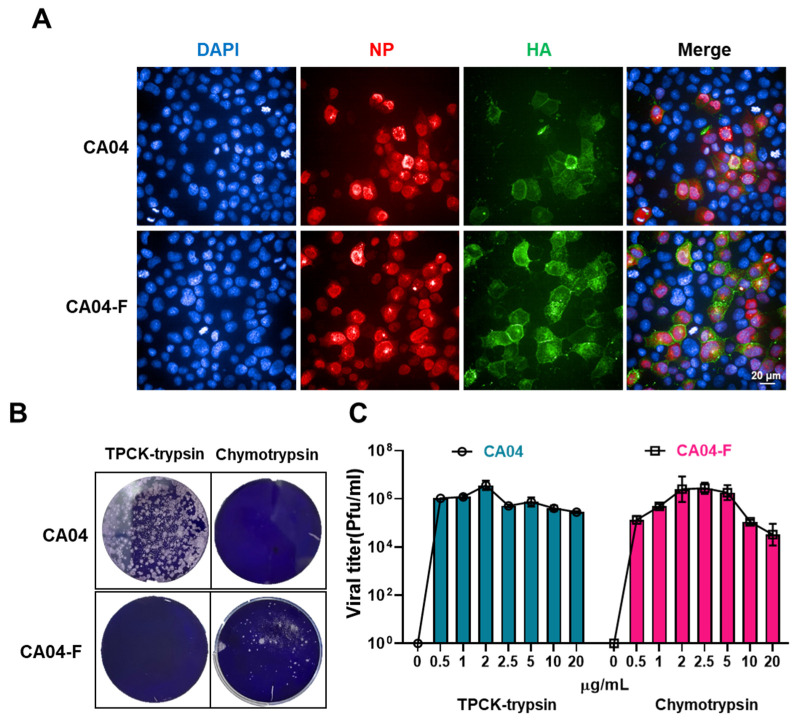
The biological characteristics of CA04-F in vitro. (**A**) The expression of HA and NP post CA04 and CA04-F infection. Blue, DAPI; red, NP; green, HA. (**B**) The specificity of CA04 and CA04-F towards TPCK trypsin and chymotrypsin. CA04 and CA04-F were subjected to plaque assays under conditions with the presence of 2.5 μg/mL TPCK trypsin or 5 μg/mL chymotrypsin. (**C**) The impact of the enzyme concentration on the viral titer in the culture supernatant. Data are presented as the mean ± s.d. values (n = 3). MDCK cells were infected with CA04 and CA04-F at an MOI of 0.1. One hour later, the medium was replaced with a medium of different enzyme concentrations, and after 48 h of incubation, the viral titer in the supernatant was measured by plaque assay.

**Figure 3 vaccines-12-00512-f003:**
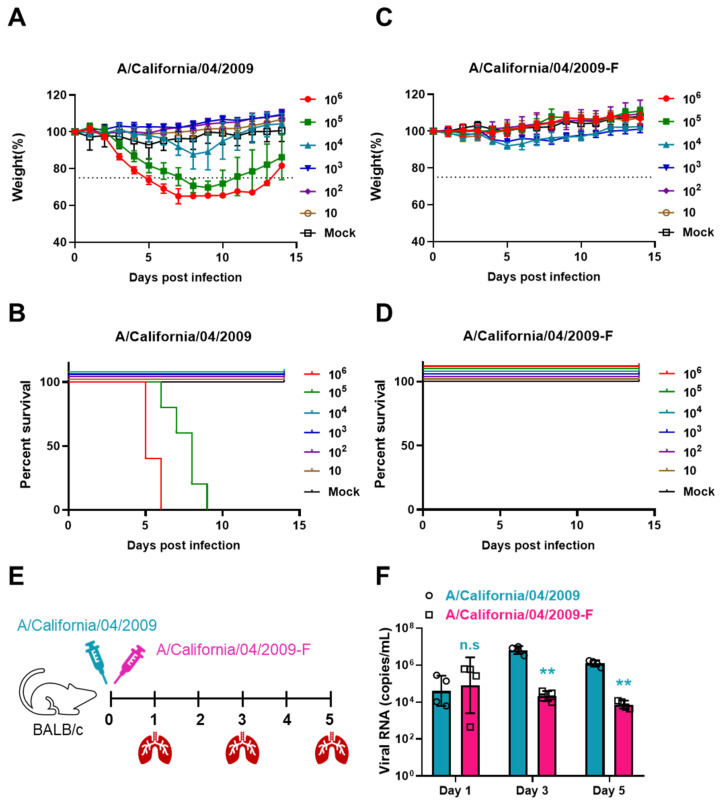
The safety of CA04-F in mice. (**A**,**B**) The weight curves (**A**) and survival rate (**B**) curves of mice were monitored post CA04 infection. 10^6^ pfu/mL CA04 was serially diluted 10 times. In total, 50 μL of the virus were administered intranasally to the mice. The weight and survival rate of the mice were monitored. When the weight of the mouse decreased by more than 25%, it was counted as dead in the survival rate statistics. Data are presented as the mean ± s.d. values (n = 5). (**C**,**D**) The weight curves (**C**) and survival rate (**D**) curves of the mice were monitored post CA04-F infection. (**E**,**F**) 50 μL 10^6^ pfu/mL CA04 and CA04-F were administered intranasally to the mice. Lung tissues were harvested on day 1, day 3, and day 5 post-virus infection to measure the viral RNA copies in the lungs through qRT-PCR. Timeline of the experimental setup for the experiment to detect viral RNA copies (**E**). Viral RNA copies in the lungs on day 1, day 3, and day 5 post-infection (**F**). Data are presented as the mean ± s.d. values (n = 4). Statistical analysis was performed by a *t* test in (**F**). n.s., no statistical significance; ** *p* < 0.01.

**Figure 4 vaccines-12-00512-f004:**
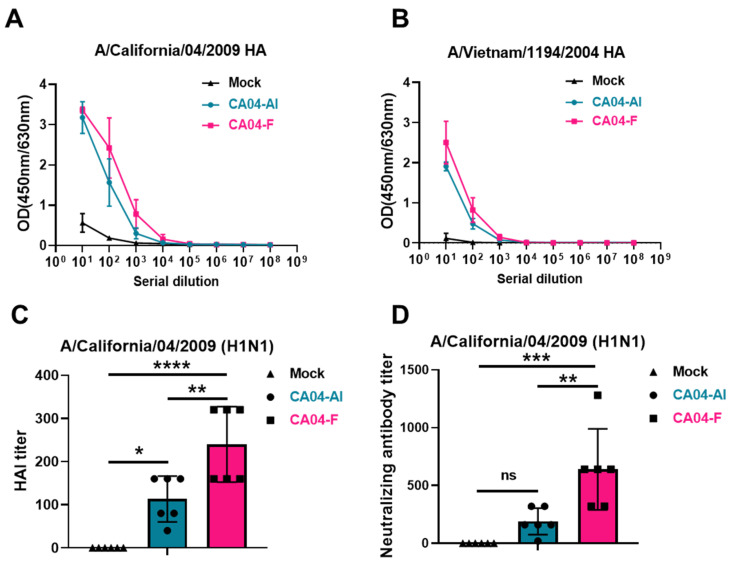
Humoral immune response in mice post mucosal immunization with CA04-F or intramuscular immunization with the mixture of inactivated CA04 and aluminum adjuvant. (**A**,**B**) The binding activity of the immunized mouse serum to A/California/04/2009 (H1N1) and A/Vietnam/1194/2004 (H5N1) HA. Data are presented as the mean ± s.d. values (n = 6). (**C**) The hemagglutination inhibition (HAI) activity of immunized mouse serum against H1N1. (**D**) The neutralization activity of immunized mouse serum against H1N1. Data are presented as the mean ± s.d. values (n = 6) in (**C**,**D**). Statistical analysis was performed by one-way ANOVA in (**C**,**D**). ns, no statistical significance; * *p* < 0.05; ** *p* < 0.01; *** *p* < 0.001; **** *p* < 0.0001. Mucosal immunization with CA04-F, CA04-F; intramuscular immunization with the mixture of inactivated CA04 and aluminum adjuvant, CA04-Al.

**Figure 5 vaccines-12-00512-f005:**
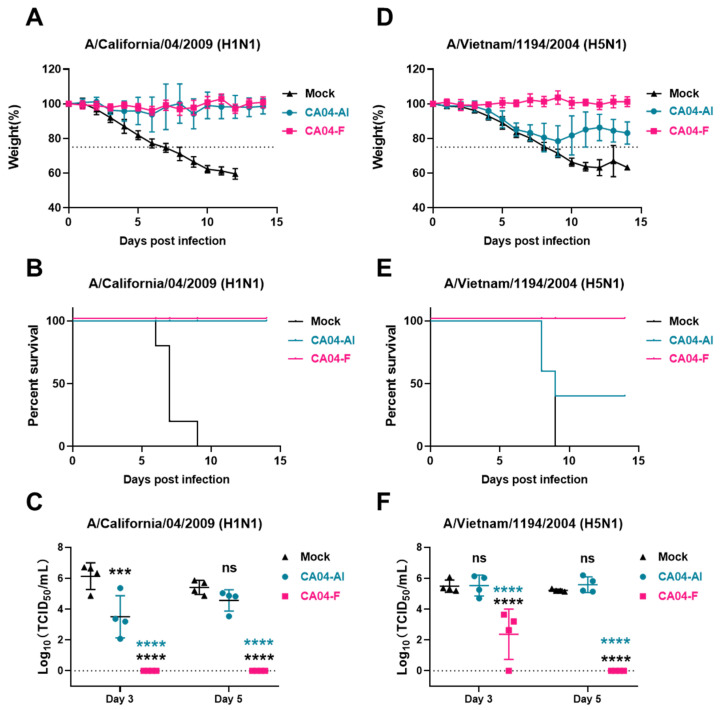
Broad-spectrum protective activity of CA04-F mucosal immunization against H1N1 and H5N1. (**A**–**C**) The protective activity of CA04-F mucosal immunization or inactivated CA04 intramuscular immunization against H1N1. Mice immunized with mock, CA04-F mucosal immunization or inactivated CA04 via intramuscular injection were challenged with a lethal dose of A/California/04/2009 to assess the effectiveness of the vaccines. The weight and survival rate of the mice were monitored, and weight curve (**A**) and survival rate curve (**B**) were plotted. Lung tissue from the mice was harvested on day 3 and day 5 post-virus infection for the detection of influenza virus titer (**C**). (**D**–**F**) The protective activity of CA04-F mucosal immunization or inactivated CA04 intramuscular immunization against H5N1. The weight and survival rate of the mice were monitored, and the weight curve (**D**) and survival rate curve (**E**) were plotted. The titers of the influenza virus in the lung tissues of mice on day 3 and day 5 post virus infection were presented (**F**). Data are presented as the mean ± s.d. values (n = 5) in (**A**,**D**). Statistical analysis was performed by two-way ANOVA in (**C**,**F**). ns, no statistical significance; *** *p* < 0.001; **** *p* < 0.0001.

## Data Availability

Data are contained within the article.

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
