# Peer review of "A Chymotrypsin-Dependent Live-Attenuated Influenza Vaccine Provides Protective Immunity against Homologous and Heterologous Viruses"

_vaccines, 2024, doi:10.3390/vaccines12050512_

Round 1

Reviewer 1 Report

Comments and Suggestions for Authors

See attached file, please.

Author Response

Specific comments:

This article is dedicated to the creation of a live attenuated influenza vaccine. It seems the authors have experience in the field of vaccine against coronavirus infection (SARS-CoV-2) and try to apply a similar approach for influenza to obtain an attenuated virus as a vaccine. A target of this research is an enzymatic cleavage site in surface viral protein hemagglutinin (HA) that have crucial role in a viral life cycle. Authors have replaced cleavage site recognized by trypsin-like proteases with another one, that is specific for chymotrypsin. This is not a new idea. Earlier for this purpose HA cleavage site was replaced with a structure recognized by elastase. These works are cited in the paper [see 33, 34]. Despite the fact that the introduction does not correspond to the topic, the experimental part of this work is well done. The modern techniques were applied to create a chymotrypsindependent influenza virus by revers genetics and to test its properties in cell culture (immunofluorescence and plaque assay) and in mice. The Discussion is written better than the Introduction. Here authors explain why the elastasedependent influenza virus created by other researchers is not optimal variant for live attenuated influenza vaccine [33, 34]. However, in this work, they have applied the same modification strategy to design the chymotrypsin-dependent virus, which also may have its drawbacks. All enzymes mentioned here trypsin, elastase, and chymotrypsin belong to the same class of serine proteases that are produced in mammalian. Therefore, some organs or tissues possessing one of these enzymes are potentially vulnerable to the modified influenza virus. But this is only my opinion. The authors may have another plan for future research. Main idea of this research includes obtaining attenuated virus due to cleavage site modification. The authors suggest this approach to create vaccine for other viruses such as coronaviruses and herpesviruses. (Lines 354-357). I think, this is the theme for readers to discuss. My objections, suggestions, and remarks concerning the description of methods, reagents, and the results are given below.

Remarks and suggestions Introduction

Comment 1: Rewrite Introduction and revise references. Some references contain information and data that are outdated correspond to the topic (ref. 4, phylogenetic groups of hemagglutinins). Some information was wrong interpreted during quoting (ref. 5, 9, 10, 21). For example, in lines 68-70 is written: continuous viral adaptive passage [21], truncation and deletion of the NS1 gene [22], and cold adaptation modifications have all been effective in reducing the virulence The reference [21] contains contrary data. The mouse-adapted H9N2 virus was derived from series of sequential lung-to-lung passages of the wild-type virus in mice and became more virulent but not attenuated. You can find many articles, where mouse-adapted influenza viruses were more virulent than their precursor.

Response: We express our heartfelt gratitude for your insightful comments. Your feedback is immensely important to us. In response, we have conducted a thorough re-examination of our cited references and made the following adjustments based on the issues you raised:

  1. Regarding the problem you pointed out with Reference 4, we realized we had not updated the types of influenza HA in a timely manner. We have now added a new reference and revised the main text, changing the number of HA types from 18 to 19.
  2. Reference 5 may have lesser relevance to the topic of our study. Therefore, we have decided to remove it.
  3. For Reference 9, we added a cut-off time to make the data more reasonable and accurate.
  4. Regarding Reference 11, we have updated the reference section by replacing it with a more appropriate reference.
  5. As for Reference 21, it was a minor mistake on our part for incorporating the incorrect reference during the writing process. We have now corrected this by updating it to the correct source.

We deeply admire your rigorous scientific spirit and are sincerely appreciative of you pointing out our mistakes. This has significantly contributed to the scientific integrity of our manuscript. Once again, thank you for your valuable comments.

Comment 2: You should include only references having direct relation to this work. The main studying object is the cleavage site of an influenza virus hemagglutinin (HA). It should be note that among all known subtypes of influenza viruses there are host species specific viruses for humans (H1-H3), aquatic birds (H1-H16), poultry (H5, H7, H9), bats (H16, H18), and other (swine, horses). Sometimes, avian influenza viruses can infect human (H5, H6, H7, H9, and H10) without transmission between humans. The various HAs may differ by features and cleavage site structure. Thus, the HA cleavage site structure of the H5 and H7 subtypes is the determinant of pathogenicity. This work deals with two strains, which belong to human (H1N1) and avian (H5N1) influenza viruses. Depending on host species the HA can be cleaved by different proteases (see ref. below). This information is partially present in the article, as well as a description of the role of the HA in the viral life cycle, but it is necessary to place the accents correctly. Other remarks Lines (L.) 19-20 phrase cleavage site of the influenza virus Lines 36-37 polymerase A (PA), polymerase B1 and 2 (PB1 and PB2), and nonstructural Lines 38-40 H16, H17, or H18 HA, and group 2 comprising H3, H4, a phylogeny of hemagglutinins in this paper? Lines 40-41. The main types that infect humans and cause disease are H1, H2, H3, H5, H6, H7, H9, and H10. Incorrect phrase. See above (remark 2). Line 45. The time period is not specified.

Useful references

  1. Alphainfluenzavirus. ViralZone: a knowledge resource to understand virus diversity (PMID:20947564, DOI:/10.1093/nar/gkq901). Web resource: https://viralzone.expasy.org/6?outline=all_by_species
  2. Böttcher-Friebertshäuser E, Garten W, Matrosovich M, Klenk HD. The hemagglutinin: a determinant of pathogenicity. Curr Top Microbiol Immunol. 2014;385:3-34. doi: 10.1007/82_2014_384.

Materials and Methods.

Line 102. Designate subtype for strains A/California/04/2009 and A/Vietnam/1194/2004 here and further. Lines 102-104, 110-112. Check the sentences, please.

Line 115. Wright abbreviation of cells is HEC-293FT as has been designated in lines 96-97. Lines 124-125. Full name of solution (phosphate buffer saline) and its abbreviation should be given at the first mentioning (line 124). The second and other, it is enough to point only abbreviation (PBS, line 125).

Response: Thanks for reviewer`s comment. We have revised the introduction and references, deleting text and corresponding references that are less relevant to the topic. Additionally, we have cited useful references recommended by the reviewers. In the manuscript, we provided a more detailed description of the influenza virus subtypes. The abbreviations have been revised in the article, including the ones mentioned by the reviewers, such as HEC-293FT and PBS.

Comment 3: The description of some methods needs additional information. Particularly, ELISA (Lines 133-136). Could you, please, specify in detail how were obtained and purified hemagglutinins of H1N1 and H5N1 viruses, or where they were produced and purchased? It is important to understand results (Fig. 4A, 4B). How did you stop reaction in ELISA? What kind of reagent was used? (L 143) What kind of units were used to express the viral titer determined by plaque assay (Line 153) and by neutralization assays (Line 201-202).

Response: Thanks for reviewer`s comment. The HA proteins of H5N1 and H1N1 were expressed using the 293 eukaryotic expression system and obtained through affinity purification with a nickel column. We have supplemented the related information in the Materials and Methods section. Please refer to Page 7 Line 8 to Line 14. In the ELISA experiments, we used 2M H2SO4 for stop reaction. Please refer to Page 9 Line 5. Plaques were counted, averaged and multiplied by the dilution factor to determine viral titer as PFU/ml. Please refer to Page 9 Line 14 to Line 15. The neutralization titer was the lowest serum concentration that was negative for hemagglutination. Please refer to Page 12 Line 8.

Comment 4: Describing experiments with animals you did not indicate lethal doses of H1N1 and H5N1 viruses for mice. Besides, what kind of viral strains (name) were used for challenge? (line 180).

Response: Thanks for reviewer`s comment. The lethal dose and the names of the strains used for challenge have been added. Please refer to Page 10 Line 22.

Comment 5: What does it mean receptor-destroying enzyme (RDE)? In lines 187-188, abbreviation HA is used for hemagglutination, that coincides with the same designation of hemagglutinin (HA). Maybe, would be better here to designate hemagglutination titer as hemagglutination units (HAU)?

Response: Thanks for reviewer`s comment. The use of receptor-destroying enzyme can reduce nonspecificity in hemagglutination assay. We have designated hemagglutination titer as hemagglutination units (HAU). Please refer to Page 11 Line 10 and Page 27 Line 6.

Comment 6: In Material and Methods I did not find appropriative reference or description of hemagglutination assay mentioned in line 219.

Response: Thanks for reviewer`s comment. We have added a description about the HA assay. Please refer to Page 11 Line 5 to Line 10.

Results

Comment 7I would like to comment some results represented in paragraph The humoral immune response activity after mucosal immunization with CA04-F (Lines 258-261, Fig. 4A,4B): HA protein of H5N1, it was found that the CA04-F mucosal immunization group showed stronger If I have right understood the ELISA was applied to detect binding between murine serum and viral hemagglutinin. The cross- reactivity of H1N1 immune murine serum with A/Vietnam/1194/2004 (H5N1) hemagglutinin is strange, therefore I ask you kindly to describe this assay in detail (see #3 above). Origin of HAs is very important. If the recombinant protein HA was not well purified and have contained part of other viral proteins like nucleoprotein (NP) you could observed nonspecific binding. What kind of materials were used as positive and negative controls? More reliable results were obtained in hemagglutination inhibition (HIA) and microneutralization assays (line 264-265) when no serum reacted with H5N1. Protective efficacy of CA04-F intranasal immunization against H1N1 and H5N1 was shown in mice (Lines 269-291, Figure 5). Partial protection against challenge by heterological virus H5N1 can be explained by humoral immune response induced by neuraminidase of the similar subtype N1, that has conservative structure. This is confirmed by higher load of viruses in lung of mice infected with heterological virus H5N1 than homological H1N1 (Fig. 5F). I wonder, how was prepared inactivated vaccine? What kind of temperature was applied and how long? As far as I know thermal inactivation leads to disruption of HA structure that can be controlled by hemagglutination assay.

Response: Thanks for reviewer`s comment. Regarding the source of HA protein, we have described it in the materials and methods section; it is expressed using the 293 eukaryotic expression system and purified through a nickel column, please refer to Page 7 Line 8 to Line 14. The purified protein was tested using HA protein-specific antibodies, confirming its good biological activity. Therefore, we used this batch of proteins to test the binding activity of immune serum, with the serum from vehicle control group mice serving as the negative control for this experiment. We used the common method of inactivating influenza viruses, treating them at 56℃ for 30 minutes. please refer to Page 15 Line 4.

Discussion

Comment 8Line 351 PRV (RRAR), BHV-1 (RRAR), VZV (RSRR), and EBV (RRRR) Could you, please, decode these abbreviations and explain a role of gB protein in a viral life cycle of herpesviruses? Influenza A virus (Alphainfluenzavirus), mentioned above herpesviruses and coronaviruses belong to various families and have different structure but they may have similar mechanism of activation and interaction with some cellular factor during their life cycle. You should explain this and clarify why the enzymatic cleavage of definite viral proteins is important for each virus. Only after that you can discuss the cleavage site structure and its role in pathogenicity of appropriative virus.

Response: Thanks for reviewer`s comment. These abbreviations represent the amino acid sequences at the cleavage sites of these viruses. To avoid confusion, we have decided to delete these sequences. The gB protein plays a key role in membrane fusion during the virus infection process, similar to the HA protein of the influenza virus and the spike protein of SARS-COV2, all of which are fusion proteins. Please refer to Page 18 Line 22 to Page 19 Line 2.

Reviewer 2 Report

Comments and Suggestions for Authors

Manuscript ID: Vaccines-2926277

Title: A chymotrypsin-dependent live attenuated influenza vaccine provides protective immunity against homologous and heterologous viruses.

The study conducted by He and et al., presents a novel approach to expedite the development of a live attenuated influenza vaccine. This strategy involves mutating the virus's cleavage site, which leads to a vaccine that is cleaved specifically by chymotrypsin, resulting in enhanced safety. The vaccine produced using this method demonstrated both good safety profiles and broad-spectrum protective efficacy against both homologous and heterologous influenza viruses when administered via the mucosal route. These findings underscore the vaccine's potential as a universal influenza vaccine. However, certain aspects of the manuscript require further attention before publication, despite the promising results reported in the study.

A-    Materials and methods section:

1-      For plasmid construction, there are several points that the authors need to clarify:

- Was the HA gene sequence used in this study amplified from the influenza A/California/04/2009 virus, or was it commercially synthesized after optimization?

-     The authors are advised to include a table in the manuscript containing all primer sequences used in the study. This table should include additional details such as the purpose of the primer (amplification or sequencing), the length in base pairs, the accession bank number, and any other relevant information. Additionally, it would be helpful for the authors to explain why the provided primer sequences in the manuscript contain both capital and lowercase letters.

2-   In line 82, the authors stated that they mutated the Arginine at the cleavage site to Phenylalanine. In lines 103-104, the authors mentioned mutating Arginine at position 344 to either Phenylalanine (Phe), Tyrosine (Tyr), or Tryptophan (Trp). However, it is unclear how many mutations were designed for this study. Could the authors please clarify this point?

3- Again, the authors mentioned that 'The eight genes of A/California/04/2009 (GenBank No. MN371610.1-371617.1) were cloned into the phw2000 vector.' Further details regarding this process are required for clarification.

4-  In lines 117-118, the authors stated that they used 5 μg/ml α-chymotrypsin and 2.5 μg/ml TPCK-treated trypsin to treat the cells. However, in Figure 2C, they showed different concentrations for both reagents. Could the authors please reconcile this difference?

5-  For animal experiments, there are some points that the authors need to clarify:

-  The authors mentioned the use of 70 mice and then 78 mice. They need to clarify the purpose for which these mice were used.

-   For the first 70 mice, the authors stated that the mice were divided into 14 groups with 5 mice per group. However, it is not clear what these 14 groups were for or how they were used.

- For the second group of 78 mice, the authors mentioned that the mice were divided into 3 groups with 30 mice per group. However, it is not clear how correct this division was.

- It is known that inactivated vaccines are typically administered via intramuscular injection rather than intranasal administration. However, could the authors provide their insights on how they could compare the effectiveness of two vaccines given by different routes of inoculation?

6- In line 187, the authors mentioned using an influenza virus for the HI assay. Could the authors specify which strain of influenza virus was used in this assay?

 B-  Results sections:

1- It is very crucial for the authors to confirm the safety of CA04 in mice by performing qRT-PCR to quantify the attenuated virus titers and replication kinetics of the virus in MDBK cells using samples collected from mice during the experiments.

2-  Authors recommended performing a western blot assay to determine whether the expression level of the HA protein in all mutated viruses is affected compared to the non-mutated virus. This assay can help explain the comment above.

3- There is no detail about the inactivated CA04 used in this study. Therefore, the authors are highly recommended to provide more data about this aspect.

4-  It is noteworthy that some viruses continue to replicate at day 3 post-infection in mice challenged with the Vietnam strain. Regarding the viral load in the lungs collected from immunized/challenged mice, I recommend that the authors use the qRT-PCR assay for greater specificity in measuring the viral load in the lungs.

5- In lines 223-224, the authors stated that there was "successful influenza virus generation following mutations to F and Y." However, the HA titer was close to zero in CAF-Y. Could the authors explain how they reached this conclusion?

6- Figure 2B shows that the cytopathic effect (CPE) caused by CA04-F/Chymotrypsin appears significantly less compared to the CPE caused by CA04/TPCK-trypsin. However, in Figure 2C, the titration of CA04-F/Chymotrypsin is almost the same as CA04/TPCK-trypsin at 2.5 μg/ml of Chymotrypsin. Could the authors please provide an explanation for this discrepancy?

7- Figure 2C, I suggest the authors create a line bar graph showing both CA04 and CA04-F under both conditions of TPCK-trypsin and Chymotrypsin treatment.

8- For Figure 3A and C, I suggest the authors normalize all results to the mock and consider the mock as 100%. This is because I noticed that the weight of mice inoculated with different dilutions of viruses showed higher weights that were less than the mock group. Additionally, I wonder why the authors tested a lower dilution than the lethal dose instead of testing a higher dilution than the lethal dose.

 C-  Discussion Section:

-          While the discussion section was well-written, the authors should further revise it to highlight the superior efficiency of the developed live attenuated vaccine compared to other similar vaccines. They should delve deeper into their results to compare and contrast them with existing vaccine approaches, instead of focusing solely on the application of this strategy with herpesvirus and COVID in one full paragraph (line 337-356).

Comments on the Quality of English Language

NA

Author Response

Response to Reviewer Comments on the manuscript

Manuscript ID: JECC-D-23-01200

Reviewer #1

Specific comments:

The study conducted by He and et al., presents a novel approach to expedite the development of a live attenuated influenza vaccine. This strategy involves mutating the virus's cleavage site, which leads to a vaccine that is cleaved specifically by chymotrypsin, resulting in enhanced safety. The vaccine produced using this method demonstrated both good safety profiles and broad-spectrum protective efficacy against both homologous and heterologous influenza viruses when administered via the mucosal route. These findings underscore the vaccine's potential as a universal influenza vaccine. However, certain aspects of the manuscript require further attention before publication, despite the promising results reported in the study.

A-    Materials and methods section:

Comment 1: For plasmid construction, there are several points that the authors need to clarify: Was the HA gene sequence used in this study amplified from the influenza A/California/04/2009 virus, or was it commercially synthesized after optimization? The authors are advised to include a table in the manuscript containing all primer sequences used in the study. This table should include additional details such as the purpose of the primer (amplification or sequencing), the length in base pairs, the accession bank number, and any other relevant information. Additionally, it would be helpful for the authors to explain why the provided primer sequences in the manuscript contain both capital and lowercase letters.

Response: Thanks for reviewer`s comment. The HA gene sequence used in this study was amplified from the influenza A/California/04/2009 virus. In this study, the 8 plasmids of influenza A/California/04/2009 virus originate from previous research; thus, the detailed construction steps can be obtained from the reference 21 (Science bulletin 2022, 67 (13), 1372-1387.), and a detailed description is not provided again. We have revised the primer sequence. Please refer to Page 6 Line 16 to Page 7 Line 6.

Comment 2: In line 82, the authors stated that they mutated the Arginine at the cleavage site to Phenylalanine. In lines 103-104, the authors mentioned mutating Arginine at position 344 to either Phenylalanine (Phe), Tyrosine (Tyr), or Tryptophan (Trp). However, it is unclear how many mutations were designed for this study. Could the authors please clarify this point?

Response: Thanks for reviewer`s comment. In line 82, our main goal is to narrate our successful strategy for modification. In the sections on materials and methods, as well as the results, we describe our strategy for the modification of the enzymatic cleavage site by mutating Arg into three other kinds of amino acids.

Comment 3: Again, the authors mentioned that 'The eight genes of A/California/04/2009 (GenBank No. MN371610.1-371617.1) were cloned into the phw2000 vector.' Further details regarding this process are required for clarification.

Response: Thanks for reviewer`s comment. Detailed methods for constructing the 8 gene plasmids of influenza virus can be obtained from reference 21 (Science bulletin 2022, 67 (13), 1372-1387.).

Comment 4: In lines 117-118, the authors stated that they used 5 μg/ml α-chymotrypsin and 2.5 μg/ml TPCK-treated trypsin to treat the cells. However, in Figure 2C, they showed different concentrations for both reagents. Could the authors please reconcile this difference?

Response: Thanks for reviewer`s comment. In the reverse genetics process of the virus, we used a concentration of 5 μg/ml α-chymotrypsin, ensuring the HA protein could be sufficiently cleaved by the chymotrypsin. After obtaining the reverse genetics recombinant viruses, we attempted to determine the optimal enzyme concentration and found that the best concentration was 2.5 μg/ml α-chymotrypsin. Before obtaining the recombinant virus, we did not know the optimal enzyme concentration conditions.

Comment 5: For animal experiments, there are some points that the authors need to clarify:

-  The authors mentioned the use of 70 mice and then 78 mice. They need to clarify the purpose for which these mice were used.

-   For the first 70 mice, the authors stated that the mice were divided into 14 groups with 5 mice per group. However, it is not clear what these 14 groups were for or how they were used.

- For the second group of 78 mice, the authors mentioned that the mice were divided into 3 groups with 30 mice per group. However, it is not clear how correct this division was.

Response: Thanks for reviewer`s comment. 70 mice were divided into 14 groups, with 5 mice per group, to assess the safety of A/California/04/2009-F mutation (CA04-F) and A/California/04/2009 (CA04). The related data were presented in Figure 3.

About 78 mice, there was a mistake. 78 mice were divided into 3 groups, with 26 mice per group. In each group, 13 mice were used to challenge with A/Califorina/04/2009 (H1N1) and A/Vietnam/1194/2004 (H5N1), respectively. Five mice per group were monitored for body weight and survival rate, and the remaining eight were used for lung viral titer determination, with the lungs of four mice being collected on day 3 and day 5 post-virus challenge, respectively. Please refer to Page 10.

- It is known that inactivated vaccines are typically administered via intramuscular injection rather than intranasal administration. However, could the authors provide their insights on how they could compare the effectiveness of two vaccines given by different routes of inoculation?

Response: Thanks for reviewer`s comment. Inactivated vaccines administered through intramuscular injection usually generate a systemic immune response, inducing strong humoral and cellular immunity. However, when respiratory viruses invade, due to the delayed immune response, viruses can successfully infect and cause disease. In contrast, mucosal immunity can induce humoral immune responses such as sIgA and IgG, cellular immune responses, and also innate immune cells, train immunity, among other methods, directly at the respiratory mucosa. This enables an immediate antiviral function to block virus infection when viruses invade. Therefore, I believe that mucosal immunity is more effective than muscular immunity in the prevention of respiratory viral infections.

Comment 6: In line 187, the authors mentioned using an influenza virus for the HI assay. Could the authors specify which strain of influenza virus was used in this assay?

Response: Thanks for reviewer`s comment. The strains of influenza virus have been added. Please refer to Page 11 Line 14.

 B-  Results sections:

Comment 1: It is very crucial for the authors to confirm the safety of CA04 in mice by performing qRT-PCR to quantify the attenuated virus titers and replication kinetics of the virus in MDBK cells using samples collected from mice during the experiments.

Response: Thanks for reviewer`s comment. The reviewer's suggestion of utilizing RT-PCR to confirm the safety of the attenuated virus is indeed an excellent approach for monitoring the replication dynamics of the virus in vitro and in vivo. This provides us with a very important criterion for the safety assessment of attenuated viruses. Unfortunately, we have not conducted related experiments before, which means we need more time to perfect this detection method. We will develop a mature detection system in our subsequent research and use this method as an important criterion for assessing the safety of attenuated viruses in future studies.

Comment 2: Authors recommended performing a western blot assay to determine whether the expression level of the HA protein in all mutated viruses is affected compared to the non-mutated virus. This assay can help explain the comment above.

Response: Thanks for reviewer`s comment. Utilizing Western blot (WB) detection indeed helps address the comment mentioned above. Given that our team's previous research on influenza virus primarily focused on screening broad-spectrum antibodies against the head region of the influenza virus, we attempted to carry out the WB experiment using the HA-specific antibodies we had previously screened. Unfortunately, we were unable to obtain antibodies with WB activity. Therefore, we may try expressing more previously reported antibodies that recognize the stem region of the influenza virus for this experiment. However, due to various limitations such as time and materials, we regretfully acknowledge our current inability to perform this test satisfactorily. We hope to improve this detection method in our subsequent research.

Comment 3: There is no detail about the inactivated CA04 used in this study. Therefore, the authors are highly recommended to provide more data about this aspect.

Response: Thanks for reviewer`s comment. The inactivation of CA04 was carried out by treating it at 56℃ for 30 minutes. It has been added to the Materials and Methods section. Please refer to Page 15 Line 3.

Comment 4: It is noteworthy that some viruses continue to replicate at day 3 post-infection in mice challenged with the Vietnam strain. Regarding the viral load in the lungs collected from immunized/challenged mice, I recommend that the authors use the qRT-PCR assay for greater specificity in measuring the viral load in the lungs.

Response: Thanks for reviewer`s comment. The reviewer's suggestion is valuable; however, unfortunately, we do not have fresh lung tissue available for RT-PCR testing. Moreover, since we have not previously established influenza virus RT-PCR detection method, we will develop this method in our subsequent research. We hope to use this method to better control the quality of virus replication in future work.

Comment 5: In lines 223-224, the authors stated that there was "successful influenza virus generation following mutations to F and Y." However, the HA titer was close to zero in CAF-Y. Could the authors explain how they reached this conclusion?

Response: Thanks for reviewer`s comment. From Figures 1C and 1D, we can see that only the highest concentration of CA04-Y was able to cause red cell agglutination, whereas CA04-W and Mock were unable to induce red cell agglutination. The hemagglutination assay is an important indicator for detecting the generation of influenza virus. Therefore, this confirms that CA04-Y was successfully generated.

Comment 6: Figure 2B shows that the cytopathic effect (CPE) caused by CA04-F/Chymotrypsin appears significantly less compared to the CPE caused by CA04/TPCK-trypsin. However, in Figure 2C, the titration of CA04-F/Chymotrypsin is almost the same as CA04/TPCK-trypsin at 2.5 μg/ml of Chymotrypsin. Could the authors please provide an explanation for this discrepancy?

Response: Thanks for reviewer`s comment. This discrepancy may be due to the enzyme-containing medium needing to be mixed with a hot agarose solution during the virus plaque assay. During this process, the tolerance of different enzymes to temperature varies, which could lead to a certain degree of reduction in enzyme activity. This might be the reason for the differences observed between Figures 2B and 2C.

Comment 7: Figure 2C, I suggest the authors create a line bar graph showing both CA04 and CA04-F under both conditions of TPCK-trypsin and Chymotrypsin treatment.

Response: Thanks for reviewer`s comment. We have been revised the Figure 2C. Please refer to Page 29.

Comment 8: For Figure 3A and C, I suggest the authors normalize all results to the mock and consider the mock as 100%. This is because I noticed that the weight of mice inoculated with different dilutions of viruses showed higher weights that were less than the mock group. Additionally, I wonder why the authors tested a lower dilution than the lethal dose instead of testing a higher dilution than the lethal dose.

Response: Thanks for reviewer`s comment. Some groups of mice gained weight faster than the Mock group, but this change is still within the normal range and does not show significant statistical differences. The data in the Figure 3A and C were obtained by dividing the weight of mice post-infection by their initial weight before infection and then converting it into a percentage. As shown in Figure 2, under optimal enzyme concentration conditions, the titers of the two viruses were still at 106 pfu/mL, which limited our ability to conduct experiments with higher doses of virus infection. Moreover, our infection dose was higher than the lethal dose because it was clear that all mice in the first two high-dose groups of CA04 were 100% deceased. However, CA04-F, due to its attenuation modification, had a 100% survival rate in its high-dose group and did not show significant weight loss, confirming its good safety profile.

C-  Discussion Section:

Comment 1:  While the discussion section was well-written, the authors should further revise it to highlight the superior efficiency of the developed live attenuated vaccine compared to other similar vaccines. They should delve deeper into their results to compare and contrast them with existing vaccine approaches, instead of focusing solely on the application of this strategy with herpesvirus and COVID in one full paragraph (line 337-356).

Response: Thanks for reviewer`s comment. We have revised the discussion section, and added “Mucosal immunization might be one of the most effective methods to prevent respiratory virus infections. Traditional intramuscular injection vaccines can induce strong systemic humoral and cellular immune responses after immunization. When viruses invade, the delayed activation of the immune system provides an opportunity for the virus to infect and cause disease. However, respiratory mucosal immunity can generate robust humoral and cellular immunity at the respiratory mucosa. When viruses invade, they can be directly blocked or eliminated at the respiratory mucosa, offering stronger protective efficacy.” into the discussion section. Please refer to Page 20 Line 1 to Line 8.

Round 2

Reviewer 2 Report

Comments and Suggestions for Authors

NA

Comments on the Quality of English Language

NA

Round 3

Reviewer 2 Report

Comments and Suggestions for Authors

NA